# Risk factors for disruptive behaviours: protocol for a systematic review and meta-analysis of quasi-experimental evidence

Lucy Karwatowska  ,[1] Simon Russell,[1] Francesca Solmi  ,[2] Bianca Lucia De Stavola,[1] Sara Jaffe,[3] Jean-Baptiste Pingault,[4,5] Essi Viding[4]

[1]Great Ormond Street Institute of Child Health, University College London, London, UK
[2]Division of Psychiatry, University College London, London, UK
[3]Department of Psychology, University of Pennsylvania, Philadelphia, Pennsylvania, USA
[4]Department of Clinical, Educational and Health Psychology, University College London, London, UK
[5]Social, Genetic, and Developmental Psychiatry, King's College London, London, UK

**Correspondence to**
Ms Lucy Karwatowska;
lucy.karwatowska.18@ucl.ac.uk

## ABSTRACT

**Introduction** Disruptive behaviour disorders, including oppositional defiant disorder and conduct disorder, are a common set of diagnoses in childhood and adolescence, with global estimates of 5.7%, 3.6% and 2.1% for any disruptive disorder, oppositional defiant disorder and conduct disorder, respectively. There are high economic and social costs associated with disruptive behaviours and the prevalence of these disorders has increased in recent years. As such, disruptive behaviours represent an escalating major public health concern and it is important to understand what factors may influence the risk of these behaviours. Such research would inform interventions that aim to prevent the development of disruptive behaviours. The current review will identify the most stringent evidence of putative risk factors for disruptive behaviour from quasi-experimental studies, which enable stronger causal inference.

**Methods and analysis** The review will be carried out according to Preferred Reporting Items for Systematic Reviews and Meta-Analyses (PRISMA) guidelines. An electronic search of references published between 1 January 1980 and 1 March 2020 will be conducted using Medline, Embase, PsycINFO and Web of Science. Initial abstract and title screening, full-text screening and data extraction will be completed independently by two reviewers using Evidence for Policy and Practice Information (EPPI)-Reviewer 4 software. Quasi-experimental studies in the English language examining the association between any putative risk factor and a clearly defined measure of disruptive behaviour (eg, a validated questionnaire measure) will be included. We will conduct meta-analyses if we can pool a minimum of three similar studies with the same or similar exposures and outcomes.

**Ethics and dissemination** The proposed review does not require ethical approval. The results will help to identify risk factors for which there is strong evidence of causal effects on disruptive behaviours and also highlight potential risk factors that require further research. The findings will be disseminated via publication in a peer-reviewed scientific journal and through presentations at international meetings and conferences.

**PROSPERO registration number** CRD42020169313.

### Strengths and limitations of this study

► The strengths of the current study include systematically evaluating quasi-experimental research, which aims to estimate causal effects, of a wide range of risk factors for a variety of disruptive behaviours.
► The risk factors will not be selected *a priori* allowing any risk factor to be included in the review and the quality of evidence will be assessed to provide confidence in the effect estimates.
► The present study will be the first to synthesise quasi-experimental evidence for disruptive behaviours and will inform future research of the most probable risk factors associated with the development of these disorders.
► The current review will focus on quasi-experimental, as opposed to experimental, methods which, despite being more robust than other non-experimental methods, are not immune to bias.
► Bias that may be introduced through solely selecting articles published in the English language will be lessened by including studies that are conducted in any country as long as the full article is translated into English.

## BACKGROUND

Individuals who display disruptive behaviours in childhood (including conduct problems, conduct disorder and oppositional defiant disorder) engage in a range of repetitive and troublesome behaviours, such as lying, fighting and stealing. Disruptive behaviour disorders are estimated to affect 5.7% of children and adolescents globally,[1] with prevalence estimates increasing in recent years.[2 3] Disruptive behaviours place a considerable economic, social and emotional burden on society[4 5] and are therefore a major public health concern. There has been a long-standing interest in understanding what factors increase (or decrease) the risk of these behaviours. Despite the large numbers of studies that have investigated this question,

**BMJ**

much of the existing evidence draws on research that has used unrepresentative samples (ie, high-risk or clinical samples), has explored a restricted number of risk factors and/or has only focused on identifying associations between putative risk factors and outcomes.[6 7] In comparison, quasi-experimental studies can address most of these limitations: they often use large, representative samples, they are able to investigate risk factors for which classical randomised control trials are unethical, impractical or too costly, and, under certain assumptions, they can produce causal estimates.[8] Given the high societal cost of disruptive behaviours it is important that research attempts to identify the causal, as opposed to correlational, risk factors associated with these behaviours. Such research would inform efforts to design more effective interventions that aim to prevent the development of disruptive behaviour and its associated adverse long-term outcomes.

To date, only one narrative review of quasi-experimental evidence for risk factors for antisocial behaviour, a common symptom of disruptive behaviour disorders, has been published.[9] The authors concluded that there was evidence that harsh parental discipline, maltreatment, parental divorce, adolescent motherhood, maternal depression, parental antisocial behaviour, peer deviance and poverty had causal effects on antisocial behaviour. However, there was no evidence of causal effects for smoking during pregnancy, paternal depression, parental alcohol use or neighbourhood disadvantage. The review was limited to eight categories of risk factors and seven types of quasi-experimental design as, at the time, an insufficient number of existing studies had included other types of risk factors and/or quasi-experimental designs to warrant inclusion in the review. Therefore the review did not include many risk factors that have been proposed to have causal effects on disruptive behaviour, including father absence, domestic violence and bullying victimisation. It also did not include specific quasi-experimental designs, such as regression discontinuity, interrupted time series or instrumental variable analyses. However, in recent years the number of quasi-experimental studies has grown quickly and the literature on risk factors for disruptive behaviours has also increased considerably.

The aim of the current review is to identify all existing quasi-experimental studies of putative risk factors for disruptive behaviours. We will consider an inclusive range of outcomes for disruptive behaviour including diagnostic (eg, conduct disorder, oppositional defiant disorder and antisocial personality disorder) and continuous measures (eg, conduct problems), as opposed to general symptoms (eg, antisocial behaviour, delinquency). Furthermore, risk factors will not be selected *a priori*, thereby allowing any putative risk factor for disruptive behaviour to be included in the review. We will use a broad definition of quasi-experimental studies: whereby we consider all studies that aim to estimate causal effects

between risk factors and outcomes as quasi-experimental. Hence as well as including quasi-experimental study designs (eg, natural experiments, twin studies) we will also include studies where the propensity for exposure to a particular risk factor is controlled for by using analytical methods, leading to estimates of causal effects under certain assumptions (online supplementary appendix A). Given the increase in quasi-experimental designs and the use of causal inference methods in recent years, we anticipate that there will be a sufficient number of studies to perform meta-analyses. By systematically combining and summarising all relevant literature, the current review will aim to:

1. Identify risk factors for disruptive behaviours from quasi-experimental designs that provide more stringent evidencefor causal inference.
2. Examine whether the results from these studies indicate evidence of causal effects.
3. Establish whether the results vary by:
   i. Outcome (ie, conduct disorder, conduct problems, oppositional defiant disorder or antisocial personality disorder);
   ii. Sex;
   iii. Study design (eg, natural experiment);
   iv. Analytical methods (eg, g-methods); and/or
   v. Data quality (eg, bias induced by the rater assessing the exposure while knowing the outcome).

## METHODS
The current protocol has been registered with the PROSPERO database (CRD42020169313)and has been reported using the PRISMA-Protocols guidelines[10] (online supplementary appendix B). Any amendments to the protocol will be made through PROSPERO.

### Definition of key terms
#### Quasi-experimental studies
Quasi-experimental studies assess causality by aiming to replicate the counterfactual framework, also known as the potential outcomes framework. The counterfactual framework consists of comparing hypothetical scenarios whereby the *same* individual is either exposed or unexposed to a risk factor.[11–13] Quasi-experimental studies aim to reproduce this comparison either by design (eg, natural experiments) or by employing statistical methods (eg, g-methods). These statistical methods fall into two broad categories according to whether they invoke the assumption of no unmeasured confounding (eg, g-methods) or exploit the presence of instrumental variables (eg, Mendelian randomisation).

#### Disruptive behaviours
Many terms describing disruptive behaviours are used interchangeably in the literature. For the purposes of the current review, we have operationalised the following definitions in order to distinguish between key terms (online supplementary appendix A):

1. In childhood:
   i. Conduct disorders: a formal diagnosis whereby an individual displays repetitive and persistent patterns of antisocial, aggressive or defiant behaviour that amounts to significant and persistent violations of age-appropriate social expectations[14] and are diagnosable as defined by the Diagnostic and Statistical Manual of Mental Disorders (DSM-5).[15]
   ii. Conduct problems: a term to describe a range of repetitive and disruptive behaviours, such as lying, fighting and stealing that do not necessarily meet the threshold for a diagnosis of conduct disorder.
   iii. Oppositional defiant disorder: a formal diagnosis whereby an individual exhibits defiant and disobedient behaviour towards others as opposed to conduct disorder, whereby behaviours violate the rights of others and/or societal expectations.
2. In adulthood:
   i. Antisocial personality disorder: a diagnosis which involves a life-long pattern of antisocial behaviour as well as irritability and remorselessness. In the DSM-5, a diagnosis of antisocial personality disorder involves exhibiting conduct disorder in childhood.

## Eligibility criteria

Studies meeting *all* of the following criteria will be included in the review:

▶ The study must only include human participants, that is, studies of non-human animals will not be included.
▶ The study must include at least one clearly defined measure of a risk factor and at least one clearly defined measure of disruptive behaviour, which will ensure consistency in the definition of these variables. The risk factor must have occurred, but not necessarily have been measured, before the outcome.
▶ Effect sizes must be reported, or there must be enough numerical information to calculate effect sizes. Studies that do not meet this eligibility criterion but only report *p* values or qualitative findings (eg, 'not significant' or the direction of findings) and where additional data cannot be obtained on request will be described separately.
▶ The study may be conducted in any country but must be published in English. Language restriction was chosen not only for practical reasons but also because the majority of research on disruptive behaviours has been conducted in English speaking countries.[7]
▶ The study must be published between 1980 and 2020. Date restriction was chosen to maintain a level of consistency in the definition of disruptive behaviours.
▶ The study must use a quasi-experimental design or analysis. The online supplementary appendix A includes a table of definitions for the quasi-experimental terms that will be used in the current review, which has been informed and adapted from the causal inference literature.[11–23]

Studies meeting *any* of the following criteria will be excluded in the review:

▶ The study does not meet the above inclusion criteria.
▶ The study is a case report, clinical trial, editorial, letter to the editor, systematic review or meta-analysis.
▶ The study used populations selected on participant physical health problems (eg, cancer, seizures, surgery, low gestational age, etc). Symptoms of disruptive behaviour (eg, antisocial behaviour) can be exhibited by people with physical health problems. However, this is not the focus of the current review's research question.
▶ The study used populations selected *solely* on other diagnosed developmental disorders (eg, language disorders, learning disorders, motor disorders, autism spectrum disorders, etc) or mental health diagnoses (eg, schizophrenia, depression, bipolar, etc). As above, disruptive behaviour can share symptomology with other developmental and mental health disorders but these go beyond the scope of the present review.

## Search strategy

An electronic search will be conducted to identify all relevant studies. Databases will be searched from 1 January 1980 until 1 March 2020. The reference list of eligible full texts will also be screened to identify additional articles.

## Databases

The selection of electronic databases was made either due to the database being relevant for the current review's research question, eg, PsycINFO, and/or because the database is frequently used in literature searches, eg, Medline. We included the Web of Science Core Collection database as eight articles that were included in the review conducted by Jaffee and colleagues were not available in the Ovid databases but were available in Web of Science. As such, we will systematically search the following databases:

1. Ovid:
   i. Medline In-Process and Other Non-Indexed Citations and Daily.
   ii. Embase.
   iii. PsycINFO.
2. Web of Science Core Collection:
   i. Science Citation Index Expanded (SCI-EXPANDED)- 1900-present.
   ii. Social Sciences Citation Index (SSCI)-1900-present.
   iii. Arts and Humanities Citation Index (A&HCI)- 1975-present.
   iv. Conference Proceedings Citation Index—Science (CPCI-S)- 1900-present.
   v. Conference Proceedings Citation Index—Social Science and Humanities (CPCI-SSH)-1900-present.
   vi. Book Citation Index—Science (BKCI-S)- 2005-present.
   vii. Book Citation Index—Social Sciences and Humanities (BKCI-SSH)- 2005-present.

viii. Emerging Sources Citation Index (ESCI)- 2015-present.

## Search terms

The search terms used for quasi-experimental studies were adapted from a paper by Glanville *et al*[24] to include genetically informed causal inference methods, such as twin designs and Mendelian randomisation, in line with changes in recent literature. The search terms for disruptive behaviours were selected to include diagnostic terms, for example, conduct disorder and antisocial personality disorder. We decided not to include terms for symptoms associated with disruptive behaviour, such as antisocial behaviour and delinquency, as they are not specific to disruptive behaviour. As mentioned above, these behaviours are also exhibited in other disorders, including autism spectrum disorder, which are not relevant to the current review's research aims. Table 1 displays the keywords that will be used in the current review and table 2 shows the techniques that will be employed in the database searches.

## Study selection

Citations will be imported into Evidence for Policy and Practice Information (EPPI)-Reviewer 4,[25] a data management software. EPPI-4 includes a machine learning process which will reduce the time taken to screen titles and abstracts by prioritising unscreened articles based on the reviewers' previous screening decisions; specifically, the EPPI-4 software assesses the frequency of words in the inclusion compared with the exclusion categories. Two independent reviewers (LK and FS) will complete the initial screening of abstract and titles using EPPI-4. The consistency between reviewers' screening decisions will be checked periodically and the list of unscreened references will be refreshed, allowing the machine learning software to prioritise unscreened items based on relevance denoted from the inclusion and exclusion codes. Once 300 references have been excluded consecutively (ie, no references have been included) by both reviewers, the remaining references will be categorised as exclude on title and abstract. In order to double-check that these references have been correctly categorised by the software, a random selection of 5% of the unscreened references will be checked by both reviewers for inclusion. If any of these references are categorised as include, the list of unscreened references will be refreshed and both reviewers will continue screening until 300 references have been excluded consecutively. This process will be repeated until a random selection of 5% of the unscreened references are confirmed as exclude by both reviewers. Any references that are categorised as include by both reviewers after screening on title and abstracts will have their full-texts screened for inclusion by two independent reviewers (LK and FS). Any uncertainties over the inclusion/exclusion of studies will be resolved by team consultation.

## Data extraction and quality assessment

Data extraction and quality assessment will be conducted by two independent reviewers (LK and SR) on any references categorised as include after the full-text screening. Any discrepancies will be resolved through discussion and any missing data will be requested from study authors. A data extraction tool will be created (online supplementary appendix C) and piloted on a random selection of ten references included after the full-text screening. The data extraction tool will be reviewed and modified if necessary, after group discussion. The following information will be extracted from the studies using a data extraction form: study reference, project name and country, study design (eg, cohort study), participant information (eg, number, ethnicity, age at measurement), main exposure and outcome measurement features (eg, measurement tool, rater, age at measurement), confounders, additional risk factors, additional outcomes, average effect size and other relevant quantities (eg, estimate, SE, sample size, exclusions, attrition). The online supplementary appendix C includes further details on the information that will be extracted. Checklists for assessment of study quality will be adapted from the Risk of Bias in Non-randomised Studies of Exposures checklist (online supplementary appendix D).

## Strategy for data synthesis

We will provide a description of the data extracted from the selected studies, which will include the study designs used, participant characteristics, exposures and outcomes studied, measurement tools/raters and statistical methods. The magnitude of effects and quality of the studies will be described and critiqued, with potential avenues for further research discussed.

Depending on the amount and quality of information provided in the included studies and whether evidence on exposures and outcomes is available, we will conduct either qualitative syntheses (ie, a systematic review) or quantitative syntheses (ie, a meta-analysis). We anticipate that we will have a sufficient number of studies to be able to perform meta-analyses leading to pooled effect sizes for the association between risk factors and disruptive behaviour. We will only conduct meta-analyses if a minimum of three studies report effect estimates on a particular risk factor and a particular outcome that are sufficiently homogenous for their meta-analysis to lead to sensible summary estimates.

When appropriate, we will also stratify the results by predefined categories (see below). Random-effects meta-analyses will be used to account for likely study heterogeneities, with the resulting pooled estimates reported together with measures of their dispersion. The $I^2$ statistic will be used to quantify heterogeneity and tau-squared ($\tau^2$) will be used to indicate the extent of between-study variance. Meta-analyses of effects for binary outcomes that are measured on different scales (eg, odds ratios and risk ratios) will be considered only if events are rare and

**Table 1** Search strategy

**Search terms**

| Database | MeSH terms |
|---|---|
| Ovid Medline | **Quasi-experimental studies**—causality/; adoption/; child, adopted/; exp twins/; twin study/; propensity score/; siblings/; interrupted time series analysis/; mendelian randomization analysis/; ecological momentary assessment/; fertilization in vitro/; controlled before-after studies/; fuzzy logic/.<br>**Disruptive behaviours**—conduct disorder/; "attention deficit and disruptive behavior disorders"/; antisocial personality disorder/. |
| Ovid Embase | **Quasi-experimental studies**—causality/; causal attribution/; causal modelling/; quasi experimental study/; adopted child/; adoption/; twins/; twin study/; propensity score/; sibling/; sibling relation/; instrumental variable analysis/; time series analysis/; mendelian randomization analysis/; ecological momentary assessment/; in vitro fertilization/; exogenous variable/; fuzzy logic/; fuzzy system/; maximum likelihood method/.<br>**Disruptive behaviours**—conduct disorder/; oppositional defiant disorder/; disruptive behavior/; antisocial personality disorder/; psychopathy/. |
| Ovid PsycINFO | **Quasi-experimental studies**—exp causality/; exp causal analysis/; exp quasi experimental methods/; adopted children/; "adoption (child)"/; adoptive parents/; twins/; exp siblings/; exp time series/; exp ecological momentary assessment/; reproductive technology/; counterfactual thinking/; fuzzy logic/; fuzzy set theory/; exp maximum likelihood/.<br>**Disruptive behaviours**—exp conduct disorder/; exp oppositional defiant disorder/; exp disruptive behavior disorders/; exp externalizing symptoms/; antisocial personality disorder/; psychopathy/. |
| Web of Science | **Quasi-experimental studies**—Not applicable.<br>**Disruptive behaviours**—Not applicable. |

**Free text search terms**

| Concept 1—<br>Quasi-experimental studies | 1. Quasi-experimental studies MeSH Terms<br>2. (causal*)<br>3. ((quasiexperiment*) or (quasi experiment*))<br>4. (adopt*)<br>5. (fixed effect*)<br>6. (twin*)<br>7. (propensity score*)<br>8. (sibling*)<br>9. (regression discontinuity)<br>10. (instrumental variable*)<br>11. (interrupted time series)<br>12. (mendelian randomi?ation)<br>13. (matching stud*)<br>14. (experience sampl*)<br>15. (ecological momentary assessment*)<br>16. ((difference* in difference*) or (difference* stud*))<br>17. (in vitro fertili?ation)<br>18. ((polygenic score*) or (polygenic risk score*))<br>19. (exogenous varia*)<br>20. (natural experiment*)<br>21. (matched control*)<br>22. (counterfactual*)<br>23. (potential outcome*)<br>24. ((balancing adj3 covariate) or (imbalance adj3 covariate) or (balanced adj3 covariate) or (imbalanced adj3 covariate))<br>25. (controlled before and after) or (controlled before after))<br>26. (inverse probability weight*)<br>27. ((doubly robust regression*) or (doubly robust estimate*))<br>28. ((selection model*) or (selectivity model*))<br>29. ((heckit model*) or (heckman sample selection*))<br>30. (selection correction*)<br>31. (two stage residual inclusion*)<br>32. ((sharp design*) or (fuzzy design*))<br>33. (forcing variable*)<br>34. (full information maximum likelihood)<br>35. (natural control*)<br>36. 1 OR 2 OR 3 OR 4 OR 5 OR 6 OR 7 OR 8 OR 9 OR 10 OR 11 OR 12 OR 13 OR 14 OR 15 OR 16 OR 17 OR 18 OR 19 OR 20 OR 21 OR 22 OR 23 OR 24 OR 25 OR 26 OR 27 OR 28 OR 29 OR 30 OR 31 OR 32 OR 33 OR 34 OR 35 |

**Table 1** Continued

**Search terms**

| Concept 2— Disruptive behaviours | 37. Disruptive behaviours MeSH Terms |
| | 38. (conduct problem*) |
| | 39. (conduct disorder*) |
| | 40. (oppositional defian*) |
| | 41. (disruptive behaviour*) or (disruptive behaviour*) |
| | 42. (externali?ing) |
| | 43. (antisocial personalit*) or (anti social personalit*) |
| | 44. (dissocial personalit*) |
| | 45. ((psychopathic) or (psychopathy) or (psychopath) or (psychopaths)) |
| | 46. 37 OR 38 OR 39 OR 40 OR 41 OR 42 OR 43 OR 44 OR 45 |
| | 47. 36 AND 46 |
| | 48. Limit to English Language |
| | 49. Human not animal |
| | 50. Not review or meta-analysis or case-report |
| | 51. Limit to 1980—Current |

**Table 2** Tools and techniques for searching databases

| Technique and description | Command | Example |
| --- | --- | --- |
| **All known synonyms and spellings of key words** | | *Doubly robust regression* could also be referred to as *doubly robust estimate* |
| **Replace up to one character in the word**—allows alternative spellings to be included. | ? | *Mendelian randomi?ation* would include *Mendelian randomisation* and *Mendelian randomization* |
| **Truncation command**—used to acknowledge and capture alternative endings to words. | * | *Inverse probability weight** would additionally search for *inverse probability weights* and *inverse probability weighting* |
| **Boolean logic operators**—used to either: (a) identify **results with at least one of the search terms present**; and, (b) to combine results of different **search terms**. | "OR" "AND" | *Conduct problem* OR conduct disorder** would retrieve articles that have *either* terms. *Conduct problem* AND causal** would only retrieve articles that have *both* terms. |
| **Proximity operators**—used to identify words within **a specified distance of each other.** | adj3 | *Balancing adj3 covariate* would identify articles whereby "balancing" and "covariate" are within three words of each other. |

after appropriate transformations. The Metafor package in R[26] will be used to conduct the analyses.

### Analysis of subgroups

When sufficient data are available, meta-analyses can be used to assess the specificity of pooled effects by examining whether the effects vary across pre-specified 'subgroups'. Subgroups will be defined according to a variety of participant characteristics (eg, outcome, sex and age of onset) and study characteristics (eg, study design, analytical method, in particular level of confounder adjustment and data quality) as follows: outcomes will be categorised as either conduct problems, conduct disorder, oppositional defiant disorder or anti-social personality disorder; sex will be categorised as male or female; age of onset will either be categorised from studies providing retrospective information on continuous (eg, months, years) or categorical data (eg, before 5 years, 5–10 years, etc); study design will be grouped according to subtypes (eg, natural experiment, twin study; online supplementary appendix A); the statistical methods used for estimation will be grouped according to the assumptions invoked for causal interpretation (eg, no unmeasured confounding or exploration of

instrumental variables; online supplementary appendix A); data quality will be categorised depending on the results from the data quality assessment (see below). The direct examination of heterogeneities across subgroups will be decided depending on the information provided by the studies included in the meta-analysis.

### Confidence in cumulative evidence

The quality of the evidence will be judged independently by two reviewers (LK and SR) using the Grading of Recommendations Assessment, Development and Evaluation scale,[27] with special consideration of recent guidelines for prognostic studies which share some similarities with quasi-experimental methods.[28] Evidence quality assessment will be performed for each risk factor and for each outcome. Five domains will be considered (risk of bias, consistency, directness, precision and publication bias) and the quality will be adjudicated as high, moderate, low or very low. If there is a discrepancy between reviewers, it will be resolved by reviewer discussion or by consultation with the team as needed. The impact of publication bias and/or unequal reporting of quantitative evidence will be examined using funnel plots.

## DISCUSSION

To our knowledge, the proposed review will be the first to systematically evaluate the existing evidence of causal effects between risk factors and disruptive behaviours. The results are expected to identify the most probable risk factors for disruptive behaviours and highlight potential risk factors that could be candidates for future research.

## ETHICS AND DISSEMINATION

Ethical approval is not required for this review as it will synthesise data from existing studies. A manuscript detailing the results will be submitted for publication in a peer-reviewed scientific journal and presented according to the PRISMA guidelines. The results of the review will also be disseminated through presentations at international meetings and conferences.

**Acknowledgements** The UCL Great Ormond Street Institute of Child Health (UCL GOS ICH) receives a proportion of funding from the Department of Health's National Institute for Health Research Biomedical Research Centre funding scheme. As LK, SR and BDS work at UCL GOS ICH, this research has benefited from funding awarded to the NIHR Great Ormond Street Hospital Biomedical Research Centre.

**Contributors** LK designed the study and developed the review questions, with guidance from BLDS, J-BP, EV and SRJ. LK drafted and registered the protocol. All authors contributed to revising the protocol manuscript. LK and FS will independently complete the initial screen. LK and SR will independently complete the data extraction and quality assessment. LK will conduct the meta-analyses. LK will write the first manuscript draft and all authors will read and approve the final version of the manuscript. LK will be the guarantor of the review.

**Funding** LK is supported by a PhD studentship from the Economic and Social Research Council and the Biotechnology and Biological Sciences Research Council (ES/P000347/1). J-BP is supported by the Medical Research Foundation 2018 Emerging Leaders Prize in Adolescent Mental Health.

**Competing interests** None declared.

**Patient and public involvement** Patients and/or the public were not involved in the design, or conduct, or reporting, or dissemination plans of this research.

**Patient consent for publication** Not required.

**Provenance and peer review** Not commissioned; externally peer reviewed.

**ORCID iDs**
Lucy Karwatowska http://orcid.org/0000-0002-6519-5190
Francesca Solmi http://orcid.org/0000-0003-0219-9503

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
