## [Reviewer comments · BMJ Open]

ARTICLE DETAILS

TITLE (PROVISIONAL)	Risk factors for disruptive behaviours: protocol for a systematic review and meta-analysis of quasi-experimental evidence.
AUTHORS	Karwatowska, Lucy; Russell, Simon; Solmi, Francesca; De Stavola, Bianca; Jaffee, Sara; Pingault, Jean-Baptiste; Viding, Essi

VERSION 1 – REVIEW

REVIEWER	Jennifer Pillay Alberta Research Centre for Health Evidence, University of Alberta, Canada
REVIEW RETURNED	26-Apr-2020

GENERAL COMMENTS	Thank you for the opportunity to review this protocol. This protocol is well written and meets most of the criteria in the PRISMA checklist, although I have a few suggestions for the authors to consider that may improve the text and their methods. 1. Eligibility criteria: it would be good to clarify whether studies only reporting p values or qualitative findings (e.g "not significant") (without an effect measure or direction of effect) will be included.2. Study selection: this section has sentences related to data extraction and quality assessment that could be moved to the relevant sections. A sentence is missing about how full text selection will be undertaken (limited currently to title and abstract review). It would be very helpful to describe the machine learning process that is being used, e.g. is this just to prioritize what citations get screened first, or are some citations being excluded by the program or only being reviewed by one reviewer. If not only prioritizing (where all records get reviewed by 2 reviewers) it would be good to present some empirical evidence about the program's performance statistics and details about the process (relevance predictions used; training set).3. It may be very helpful to have a piloting step for the data extraction tool and comment on this here.4. If the authors do not plan to contact authors for any possible information related to data extraction and quality assessment, they could clarify this here.5. The data analysis section would be greatly strengthened by a better description of how the authors will analyze the data when meta-analysis is not possible, and what they will do if a portion of the studies report data useable for meta-analysis but others do not (e.g. only report direction of effect or p values). Is there a threshold for the proportion of studies reporting on a particular risk factor that need to report useable data, such that the authors are confident the estimates will be a reasonably valid representation of all the studies? Bias may be introduced if a few studies report data whereas several (with potentially conflicting findings) do not, so
---

	some indication about analysis and interpretations related to this would be good. Complimenting meta-analysis with other synthesis methods could be appropriate to fully assess the evidence base. There are several analytical methods that could be used for data that cannot be used in meta-analysis (harvest plots etc; Chapter 12 in most recent version of Cochrane handbook could be referred to if the authors are less aware of these methods) and outlining any that the authors may use a priori would lend rigor to the protocol. There may be multiple statistics used by the primary studies (HRs, adjusted RRs, aORs) and details of whether and how the authors will combine these for meta-analysis would be appropriate. 6. I wonder if age of onset (childhood vs. adolescence) could be an important subgroup variable? 7. I would imagine that GRADE guidance 28 for prognostic factor research https://www.ncbi.nlm.nih.gov/pubmed/31982539 will be used to guide the assessments and this may be described to some extent (e.g. RCTs are not required for high certainty). The authors could specify for quality assessment of individual studies and for across studies (GRADE) whether they will do this for each risk factor separately.
--	---

REVIEWER	Seena Fazel University of Oxford, UK
REVIEW RETURNED	11-May-2020

GENERAL COMMENTS	Three areas need further consideration:  1. The outcomes are diagnostic categories, and the search strategy reflects this. This means that disruptive behaviours that are identified using non-diagnostic approaches, such as self-report or informant questionnaires or police data, will be missed/excluded. Without them, the review should be titled more accurately as 'risk factors for conduct disorder, conduct problems and antisocial personality disorder.'  2. The subgroup analyses are welcome - can the authors be more precise about how they plan to categorise subgroups before any analyses are done. 3. The quality assessment proposed uses GRADE. Can the authors comment on whether this has been validated for quasi-experimental studies?
---

VERSION 1 – AUTHOR RESPONSE

Reviewer(s)' Comments to Author:

Reviewer: 1

Reviewer Name: Jennifer Pillay

Institution and Country: Alberta Research Centre for Health Evidence, University of Alberta, Canada

Please state any competing interests or state 'None declared': None declared

Please leave your comments for the authors below

Thank you for the opportunity to review this protocol. This protocol is well written and meets most of the criteria in the PRISMA checklist, although I have a few suggestions for the authors to consider that may improve the text and their methods.

Author Response: Thank you for your thorough and helpful comments on our manuscript.

We have taken on board your suggestions and amended our protocol accordingly.

Reviewer Comment: 1. Eligibility criteria: it would be good to clarify whether studies only reporting p values or qualitative findings (e.g "not significant") (without an effect measure or direction of effect) will be included.

Author Response: We have clarified in our eligibility criteria that any studies that do not report effect sizes, or provide enough numerical data to calculate effect sizes, will be included in the review and described separately: "Effect sizes must be reported, or there must be enough numerical information to calculate effect sizes. Studies that do not meet this eligibility criterion but only report p values or qualitative findings (e.g. "not significant" or the direction of findings) and where additional data cannot be obtained upon request will be described separately" (lines 239-242, page 7). We have also added this to our data extraction form (Appendix B, 'Effect estimates' sub-section, page 6).

Reviewer Comment: 2. Study selection: this section has sentences related to data extraction and quality assessment that could be moved to the relevant sections. A sentence is missing about how full text selection will be undertaken (limited currently to title and abstract review). It would be very helpful to describe the machine learning process that is being used, e.g. is this just to prioritize what citations get screened first, or are some citations being excluded by the program or only being reviewed by one reviewer. If not only prioritizing (where all records get reviewed by 2 reviewers) it would be good to present some empirical evidence about the program's performance statistics and details about the process (relevance predictions used; training set).

Author Response: Thank you for this helpful suggestion. We have moved all sentences relating to our data extraction tool and the risk of bias tool from the 'Study selection' sub-section to the 'Data extraction and quality assessment' sub-section (line 352 and lines 361 – 363, page 12).

We have added a sentence about how the full text selection will be undertaken: “Any references that are categorised as include by both reviewers after screening on title and abstracts will have their full-texts screened for inclusion by two independent reviewers (LK and FS).” (lines 343 – 345, page 12).

We have also included a description of how we will use the machine learning software to prioritise what citations get screened first:

“EPPI-4 includes a machine learning process which will reduce the time taken to screen titles and abstracts by prioritising unscreened articles based on the reviewers’ previous screening decisions; specifically, the EPPI-4 software assesses the frequency of words in the inclusion compared to the exclusion categories.” (lines 320 – 322, page 11).

And also “The consistency between reviewers’ screening decisions will be checked periodically and the list of unscreened references will be refreshed, allowing the machine learning software to prioritise unscreened items based on relevance denoted from include/exclude codes. Once 300 references have been excluded consecutively (i.e. no references have been included) by both reviewers, the remaining references will be categorised as exclude on title and abstract. In order to double-check that these references have been correctly categorised by the software, a random selection of 5% of the unscreened references will be checked by both reviewers for inclusion. If any of these references are categorised as include, the list of unscreened references will be refreshed and both reviewers will continue screening until 300 references have been excluded consecutively. This process will be repeated until a random selection of 5% of the unscreened references are confirmed as exclude by both reviewers.” (lines 323 – 325, pages 11 – 12).

– 345, pages 11 – 12).

Reviewer Comment: 3. It may be very helpful to have a piloting step for the data extraction tool and comment on this here.

Author Response: Thank you very much for this suggestion. We have added in a sentence about piloting the data extraction form under the ‘Data extraction and quality assessment’ subsection: “A data extraction tool will be created (online supplementary appendix B) and piloted on a random selection of ten references included after full-text screening. The data extraction tool will be reviewed and modified if necessary after group discussion.” (lines 349 – 354, page 12).

Reviewer Comment: 4. If the authors do not plan to contact authors for any possible information related to data extraction and quality assessment, they could clarify this here.

Author Response: We will request any missing information from study authors: “Any discrepancies will be resolved through discussion and any missing data will be requested from study authors.” (lines 350 – 352, page 12).

Reviewer Comment: 5. The data analysis section would be greatly strengthened by a better description of how the authors will analyze the data when meta-analysis is not possible, and what they will do if a portion of the studies report data useable for meta-analysis but others do not (e.g. only report direction of effect or p values). Is there a threshold for the proportion of studies reporting on a particular risk factor that need to report useable data, such that the authors are confident the estimates will be a reasonably valid representation of all the studies? Bias may be introduced if a few studies report data whereas several (with potentially conflicting findings) do not, so some indication about analysis and interpretations related to this would be good. Complimenting meta-analysis with other synthesis methods could be appropriate to fully assess the evidence base. There are several analytical methods that could be used for data that cannot be used in meta-analysis (harvest plots etc; Chapter 12 in most recent version of Cochrane handbook could be referred to if the authors are less aware of these methods) and outlining any that the authors may use a priori would lend rigor to the protocol. There may be multiple statistics used by the primary studies (HRs, adjusted RRs, aORs) and details of whether and how the authors will combine these for meta-analysis would be appropriate.

Author Response: Thank you for your comment. In our experience most quasi-experimental studies report effect estimates appropriately (i.e. direction of effects and a measure that enables calculation of standard errors; e.g. CIs or p values) and we therefore do not envisage including many, if any, studies that report insufficient information to meta-analyse. However if this does happen, any studies that “only report p values or qualitative findings (e.g. “not significant” or the direction of findings) and where additional data cannot be obtained upon request will be described separately” (lines 239-242, page 7)

We will conduct a meta-analysis if we can pool a minimum of three similar studies with useable data. We have further clarified this under the ‘Strategy for data synthesis’ sub-section: “We will only conduct meta-analyses if a minimum of three studies report effect estimates on a particular risk factor and a particular outcome that are sufficiently homogenous for their meta-analysis to lead to sensible summary estimates.” (lines 382 – 384, page 13).

We have added a sentence describing how we will deal with unequal reporting of quantitative evidence: “The impact of publication bias and/or unequal reporting of quantitative evidence will be examined using funnel plots. (lines 434 – 435, page 14).

We do not anticipate that we will have to combine effect estimates on different scales as we expect most of the included studies will use continuous outcome scales and therefore the effect estimates will be on the same metric. However, we have added a sentence to our manuscript detailing when we might combine effect estimates on different scales: “Meta-analyses of effects for binary outcomes that are measured on different scales (e.g. odds ratios and risk ratios) will be considered only if events are rare and after appropriate transformations.” (lines 390 – 392, page 13).

Reviewer Comment: 6. I wonder if age of onset (childhood vs. adolescence) could be an important subgroup variable?

Author Response: Thank you for this suggestion. We will extract information about age of onset from any study included after full-text screening. We have added this to our data extraction form (Appendix B, Question D8B, page 5). If there are at least three studies that report information about the same risk factor and also the age of onset of a similar outcome we will conduct exploratory moderation analyses. We have specified this under the 'Analysis of subgroups' sub-section: "age of onset will either be categorised from studies providing retrospective information on continuous (e.g. months, years) or categorical data (e.g. before 5 years, 5 – 10 years etc)" (lines 402 – 404, page 13).

Reviewer Comment: 7. I would imagine that GRADE guidance 28 for prognostic factor research <https://eur01.safelinks.protection.outlook.com/?url=https%3A%2F%2Fwww.ncbi.nlm.nih.gov%2Fpubmed%2F31982539&data=02%7C01%7C%7Cea0403a9d49d4010606008d7fab60723%7C1faf88fea9984c5b93c9210a11d9a5c2%7C0%7C0%7C637253532033894082&sdata=WgIYZ3rq%2BtGOPYxNlzXT%2BjtQ0ujESVukKUY66fsJY%2Fo%3D&reserved=0> will be used to guide the assessments and this may be described to some extent (e.g. RCTs are not required for high certainty). The authors could specify for quality assessment of individual studies and for across studies

(GRADE) whether they will do this for each risk factor separately.

Author Response: Thank you for sharing this relevant paper with us. We have added it as a reference under the 'Confidence in cumulative evidence' sub-section: "The quality of the evidence will be judged independently by two reviewers (LK and SR) using the Grading of Recommendations Assessment, Development and Evaluation (GRADE) scale [28], with special consideration of recent guidelines for prognostic studies which share some similarities with quasi-experimental methods [29]." (lines 426 – 429, page 14). We have specified that the quality assessments (GRADE) will be conducted for each risk factor and for each outcome: "Evidence quality assessment will be performed for each risk factor and for each outcome." (lines 429 – 430, page 14).

Reviewer: 2

Reviewer Name: Seena Fazel

Institution and Country: University of Oxford, UK

Please state any competing interests or state 'None declared': None declared

Please leave your comments for the authors below

Author Response: Thank you for taking the time to review our manuscript and providing helpful suggestions for us to improve our protocol.

Three areas need further consideration:

Reviewer Comment: 1. The outcomes are diagnostic categories, and the search strategy reflects this. This means that disruptive behaviours that are identified using non-diagnostic approaches, such as self-report or informant questionnaires or police data, will be missed/excluded.

Without them, the review should be titled more accurately as 'risk factors for conduct disorder, conduct problems and antisocial personality disorder.'

Author Response: Thank you for this comment. We decided that the focus of the current review should be on diagnostic terms for disruptive behaviour in order to maintain a level of consistency in the definitions used in the included studies. We had not planned to only select studies that include clinical diagnoses as outcomes but instead we had planned to select studies that include a clearly defined measure of disruptive behaviour. Research on disruptive behaviours often include continuous outcomes based on self- or other-reported questionnaire measures, such as the Child Behaviour Checklist or the Strengths and Difficulties Questionnaire. We accept that this could have been made more explicit in our manuscript and therefore we have amended our abstract and eligibility criteria in line with this need for clarification.

Abstract: "Quasi-experimental studies in the English language which examine the association between any putative risk factor and a clearly defined measure of disruptive behaviour (e.g. a validated questionnaire measure) will be included." (lines 52 – 54, page 2).

Eligibility criteria: "The study must include at least one clearly defined measure of a risk factor and at least one clearly defined measure of disruptive behaviour, which will ensure consistency in the definition of these variables." (lines 235 – 237, page 7).

In addition, since submitting our original manuscript to the BMJ Open, we have revisited our search strategy and decided that in order to identify more studies that include sub-clinical disruptive behaviours we should include a search term for "externalising" behaviours, which is often used interchangeably with "disruptive behaviour" disorders. We have included externalising in our search strategy (Table 1, pages 9 - 10) and further definitions (Appendix D) to reflect this.

Reviewer Comment: 2. The subgroup analyses are welcome - can the authors be more precise about how they plan to categorise subgroups before any analyses are done.

Author Response: Thank you for your suggestion. We have provided further details on how we plan to categorise subgroups, given that sufficient information is provided in the included studies, under the 'Analysis of subgroups' sub-section: "When sufficient data are available, meta-analyses can be used to assess the specificity of pooled effects by examining whether the effects vary across pre-specified "subgroups". Subgroups will be defined according to a variety of participant (e.g. outcome, sex and age of onset) and study characteristics (e.g. study design, analytical method, in particular level of confounder adjustment, and data quality) as follows: outcomes will be categorised as either conduct problems, conduct disorder, oppositional defiant disorder or antisocial personality disorder; sex will be categorised as male or female; age of onset will either be categorised from studies providing retrospective information on continuous (e.g. months, years) or categorical data (e.g. before 5 years, 5 – 10 years etc); study design will be grouped according to subtypes (e.g. natural experiment, twin study; online supplementary appendix D); the statistical methods used for estimation will be grouped according to the assumptions invoked for causal interpretation (e.g. no unmeasured confounding or exploration of instrumental variables; online supplementary appendix D); data quality will be categorised depending on the results from the data quality assessment (see below). The direct examination of heterogeneities across subgroups will be decided depending on the information provided by the studies included in the meta-analysis." (lines 396 – 423, pages 13-14).

Reviewer Comment: 3. The quality assessment proposed uses GRADE. Can the authors comment on whether this has been validated for quasi-experimental studies?

Author Response: We will use the recently published GRADE guidelines for prognostic research to guide our quality assessment. Although prognostic studies are not the same as quasi-experimental studies, they share similar characteristics (e.g. large, observational studies with high certainty ratings) and so we believe that the guidance is applicable to our current review. We have referenced the new guidelines under the 'Confidence in cumulative evidence' sub-section: "The quality of the evidence will be judged independently by two reviewers (LK and SR) using the Grading of Recommendations Assessment, Development and Evaluation (GRADE) scale [28], with special consideration of recent guidelines for prognostic studies which share some similarities with quasi-experimental methods [29]." (lines 426 – 429, page 14).

Since submitting our original manuscript to the BMJ Open, we have decided to use a risk of bias tool that was created specifically for quasi-experimental studies that examine the effect of an exposure on an outcome (Risk Of Bias In Non-randomised Studies - of Exposures; ROBINS-E). We believe it is more applicable to our review question (i.e. the study of risk

factors) than the ROBINS-I, which we specified in our original manuscript and was created for quasi-experimental studies that examine the effect of an intervention on an outcome. Although the ROBINS-E has not yet been validated, it is a thorough risk of bias tool which includes relevant questions for assessing risk of bias in quasi-experimental studies; for example, time-varying confounding (Appendix C). We have amended this in our risk of bias form (Appendix C) and manuscript: "Checklists for assessment of study quality will be adapted from the Risk Of Bias In Non-randomised Studies - of Exposures (ROBINS-E) checklist (online supplementary appendix C)." (lines 361 – 363, page 12).